# Autophagy in Inflammatory Response against SARS-CoV-2

**DOI:** 10.3390/ijms24054928

**Published:** 2023-03-03

**Authors:** Roxana Resnik, Fabiana Lopez Mingorance, Francisco Rivera, Florencia Mitchell, Claudio D. Gonzalez, Maria I. Vaccaro

**Affiliations:** 1Instituto de Bioquimica y Medicina Molecular, Consejo Nacional de Investigaciones Cientificas y Tecnicas, Universidad de Buenos Aires, Buenos Aires C1425FQB, Argentina; 2Centro de Educacion Medica e Investigaciones Clinicas, Medicina Traslacional, Buenos Aires C1430EFA, Argentina

**Keywords:** COVID-19, macroautophagy, ATG proteins, mitophagy, SIRS, MOF, pyroptosis, NETosis

## Abstract

The coronavirus disease pandemic, which profoundly reshaped the world in 2019 (COVID-19), and is currently ongoing, has affected over 200 countries, caused over 500 million cumulative cases, and claimed the lives of over 6.4 million people worldwide as of August 2022. The causative agent is severe acute respiratory syndrome coronavirus 2 (SARS-CoV-2). Depicting this virus’ life cycle and pathogenic mechanisms, as well as the cellular host factors and pathways involved during infection, has great relevance for the development of therapeutic strategies. Autophagy is a catabolic process that sequesters damaged cell organelles, proteins, and external invading microbes, and delivers them to the lysosomes for degradation. Autophagy would be involved in the entry, endo, and release, as well as the transcription and translation, of the viral particles in the host cell. Secretory autophagy would also be involved in developing the thrombotic immune-inflammatory syndrome seen in a significant number of COVID-19 patients that can lead to severe illness and even death. This review aims to review the main aspects that characterize the complex and not yet fully elucidated relationship between SARS-CoV-2 infection and autophagy. It briefly describes the key concepts regarding autophagy and mentions its pro- and antiviral roles, while also noting the reciprocal effect of viral infection in autophagic pathways and their clinical aspects.

## 1. Introduction

The coronavirus disease pandemic, which profoundly reshaped the world in 2019 (COVID-19), and is currently ongoing, has affected over 200 countries, caused over 500 million cumulative cases, and claimed the lives of over 6.4 million people worldwide as of August 2022 [1]. The causative agent, severe acute respiratory syndrome coronavirus 2 (SARS-CoV-2), is a member of the Coronaviridae family of viruses, a group of positive-sense, single-stranded RNA genome viruses of approximately 30 kb in length [2]. Its genome consists of 11 genes with 11 reading frames that produce 16 non-structural proteins (NSP1 to NSP16) and 4 structural proteins, including the fusion trimeric spike (S), the envelope protein (E), the nucleocapsid protein (N), and the membrane glycoprotein (M) [3]. These reading frames generate eight accessory proteins: ORF3a, ORF3b, ORF6, ORF7a, ORF7b, ORF8a, ORF8b, and ORF9b [3,4]. NSPs are key for viral RNA replication and immune avoidance, and the accessory proteins play diverse roles, aiding in viral infection, replication, and transmission [5]. Figure 1 contains a diagram of the SARS-CoV-2 virus genome, highlighting the two open reading frames: ORF1a and ORF1b [4].

Autophagy is a catabolic process that sequesters damaged cell organelles, proteins, and external invading microbes and delivers them to the lysosomes for degradation [6,7,8,9]. It is a fundamental and evolutionarily conserved eukaryotic cellular process that has multiple effects on cell survival, homeostasis, and immunity [10]. Non-canonical autophagy describes other processes that use the autophagy molecular machinery, such as phagocytosis, inflammatory signaling, and secretion [11].

Autophagy plays an essential role in promoting RNA virus replication by inhibiting innate antivirus immune responses [12], or promoting infectivity via the autophagy-related secretion of vesicles loaded with virus [13,14,15,16].

Coronaviruses can cause respiratory infections of varying severity. Seasonal human CoVs (HCoVs) can cause mild to moderate upper respiratory tract infections with cold-like symptoms in humans [17]. On the other hand, highly pathogenic beta-CoVs have been responsible for multiple deadly outbreaks in the 21st century, including SARS-CoV (2003), the Middle East respiratory syndrome coronavirus (MERS-CoV) in 2012, and the current SARS-CoV-2 (2019) [17,18]. For the ongoing pandemic, self-isolation quarantine and vaccination are still the mainstream strategies in responding to the virus, and therapeutic strategies remain challenging. For knowledge and therapeutic purposes, depicting the virus’ life cycle and pathogenic mechanisms, as well as the cellular host factors and pathways involved during SARS-CoV-2 infection, has great relevance [19]. The viral replication process comprises mainly six steps: binding and attachment, where the virus and the host cell receptors interact—this interaction is performed by the viral protein spike, (S) and the cell host receptors, angiotensin-converting enzyme-2 (ACE2), TMPRSS2 and integrins [19]; viral entry, using the two entry pathways of membrane fusion and endocytosis—where the virus fuses with the host cell membrane via the cleavage of protein S by the serine proteases of the host cell endocytic pathway; transcription and translation, where viral proteins are translated from viral RNA; viral replication; nucleocapsid packaging, where new virion packaging at the endoplasmic reticulum (ER) and Golgi apparatus occurs; budding and egress, i.e., the release of viral particles by exocytosis, as these RNA viruses traverse the Golgi apparatus and trans-Golgi network (TGN), where envelope proteins receive additional post-translational modifications and exit the cell via lysosomal exocytosis [20].

The assembled viral components further undergo maturation in the Golgi vesicles to form the mature virion, ready to be released to the extracellular environment [21,22,23]. Gosh and colleagues found that β-coronaviruses utilize lysosomal trafficking to exit host cells, rather than other conventional secretory pathways [24].

This review aims to describe the main aspects that characterize the complex and not yet fully elucidated relationship between SARS-CoV-2 infection and autophagy. It briefly describes the key concepts regarding autophagy, including its pro- and antiviral roles, while also mentioning the reciprocal effect of viral infection in autophagic pathways and their clinical consequences. Thus, this review is relevant to the goals of developing new therapeutic targets and opportunities to treat COVID-19 and establishing better prognostic markers by describing molecules that correlate with disease severity, in order to gain a better understanding of a disease that has modified the world in such an impactful way.

## 2. Overview of Autophagy

Canonical autophagy includes three different major types: (1) microautophagy, which implies the direct sequestering of small molecules by the invagination or protrusion of the membranes of lysosomes [25]; (2) chaperone-mediated autophagy (CMA), which transfers proteins or other targeted molecules through a chaperone to the lysosome, specifically to its LAMP2A receptor; regarding this, Cuervo and colleagues led to the identification of the lysosome-associated membrane protein type 2A (LAMP-2A) as a CMA receptor [26]; LAMP2A is a single-span membrane protein with a very heavily glycosylated luminal region and a short (12 amino acids) C-terminus tail exposed on the surfaces of the lysosomes, where substrate proteins bind [27]; (3) macroautophagy, which is the third and most prevalent form, in which double-membrane vesicles called auto-phagosomes form and engulf the substrates to be transported to the lysosome, in which they then fuse and release their cargo for degradation [28,29].

The molecular machinery involved in macroautophagy is composed of evolutionarily conserved autophagy-related proteins (ATG) [7,30,31].

ATG protein complexes are sequentially recruited for autophagosome biogenesis, and, after successive steps, the autophagosome fuses with a lysosome, forming the degradation compartment called the autolysosome (Figure 2).

Autophagy is induced in mammalian cells by mTORC1 inhibition and AMPK activation, which activate autophagy when a lack of nutrients or energy is detected. The UKL1 complex, ULK1-ATG13-FIP200-ATG101, drives the pre-autophagosomal structure. Then, the class III PI3K-kinase complex, BCN1-ATG14-VPS34-VPS15, mediates membrane phosphorylation and two ubiquitin-like systems involving ATG7, ATG12, ATG10, ATG5, ATG16, and ATG3, which allows the recruitment of Atg8 family proteins, promoting membrane expansion and cargo sequestration [7,32,33]. Two transmembrane proteins are involved in this sophisticated membrane trafficking process, ATG9 and (VMP1) [34,35,36,37]. VMP1 is a well-described transmembrane protein essential for mammalian cell autophagy [7,34]. TMEM41B is a transmembrane protein recently linked to autophagy that shares a conserved VTT domain with VMP1, and, together, they belong to the conserved DedA family of half transporters [14,38]. Finally, soluble N-ethylmaleimide sensitive factor attachment protein receptors (SNAREs), such as STX17, facilitate the fusion between autophagosomes and lysosomes, forming the autolysosome structure, where cargoes are degraded and recycled into the cytosol [39]. The autophagosomal fusion is regulated by the homotypic fusion and protein sorting (HOPS) complex. Then, mTOR phosphorylates UVRAG and activates the PI3KC3-C2 complex to produce a lysosomal pool of PI3P. In this way, mTOR controls this process and regulates tubular initiation and maintenance. Macroautophagy can be non-selective or selective. In selective autophagy, one specific cellular component is recognized and sequestered in autophagosomes [40]. Examples of this mechanism are mitophagy, lipophagy, zymophagy, and xenophagy, the latter being a pathway by which the cell sequesters and degrades external invading microorganisms, including viruses, as a protective defense mechanism [41] (Figure 2).

Non-canonical autophagy describes other processes that use the autophagy molecular machinery, such as phagocytosis, inflammatory signaling, and secretion. One of the most relevant non-canonical autophagy pathways is the recently described secretory autophagy [42]. Secretory autophagy has been related to unconventional secretion, which links autophagy with an anabolic condition where cytosolic proteins lack an N-terminal signal peptide, which is the main peptide needed to undergo the conventional secretion pathway through the ER and Golgi apparatus. These are cytosolic proteins secreted from the cells to perform their biological functions. This process is associated with the canonical autophagic pathway as it shares regulation factors (ATGs), which also lead to the formation of autophagic membranes. These autophagic regulating factors include ULKs, BCN1, LC3s, and GABARAPs (analogs of the yeast Atg8) [42].

## 3. Autophagy and the Viral Replication Cycle

There is considerable evidence suggesting that autophagy could play a role in assisting with the viral replication cycle at different stages.

As is well known, the SARS-CoV-2 virus enters the host by binding its receptor-binding domain (RBD), located in the spike protein, to the ACE2 host cell receptor. Moreover, a bioinformatic analysis showed that two integrin-binding regions are present in the cytoplasmic C-terminal tail of the host ACE2 protein. Additionally, the viral S protein has an arginine–glycine–aspartic acid (RGD) motif-binding domain that links to integrins, suggesting that integrins could be used as co-receptors for viral entry [43].

Furthermore, one specific type of short linear motif (SLiM) has been found to be present in the tails of integrin β3 and ACE2. Thus, these structures from the autophagic machinery could be used to facilitate viral attachment, entry, and replication (as highlighted in the first step of Figure 3). Additionally, LC3-interacting region motifs (LIRs) have been identified in the tails of integrin β3 and ACE2. These LIRs are involved in the autophagic pathway through interaction with LC3, a marker of autophagosomes [19,44]. These LIR motifs also facilitate viral attachment, entry, and replication. Several viruses, including coronaviruses (CoVs), take advantage of cellular autophagy to facilitate their own replication. SARS-CoV-2 mediates its replication through a dependent or independent ATG5 pathway using specific double-membrane vesicles that can be considered similar to autophagosomes [45].

As previously described, the virus enters the cell via the endosomal pathway. The spike protein is composed of two subunits: one binding subunit (S1) and one fusion subunit (S2). First, the S1 subunit binds to the host cell receptor ACE2. This union induces conformational changes in protein S. Second, the S2 subunit is activated. This activation occurs through two consecutive cleavage steps: a cleavage between the S1 and S2 domains of protein S by Furin is produced, and it then undergoes further cleavage at the S2’ site, which promotes the unmasking and activation of the fusion peptide [46,47]. To activate the spike protein’s fusion potential, a second cleavage performed by the host’s proteases is required. The cleavage can occur at different stages of the virus infection cycle by different host proteases, such as Furin (convertase), TMPRSS2 (cell surface protease), and Cathepsin L (lysosomal protease) [48,49]. Cathepsin L, which links the virus cycle to the autophagy process, acts at a low pH, degrades cargo, and maintains autolysosome homeostasis and autophagic flux [50]. By cleaving the spike protein at S2’, it mediates virus membrane and autolysosome fusion, thus facilitating the release of viral RNA into the host cell. Smieszek et al., demonstrated that the inhibition of Cathepsin L could significantly reduce the entry of viruses into host cells [51]. It has also been shown that TMPRSS2, Furin, and Cathepsin L proteases have cumulative effects to activate virus entry and increase the pathogenicity of SARS-CoV-2 [52,53] (Figure 3, pathway 4).

Once the virus has entered the cell, it uses the autophagy machinery for its own benefit through viral proteins, NSPs. Recently, as shown in Figure 3, pathway 3, transmembrane proteins related to autophagy, i.e., VMP1 and TEMEM41B, have been reported as critical host factors at the early stages of viral infection [52]. VMP1 and TMEM41B both contribute to different stages of DMV formation. Ji et al., have revealed that DMV biogenesis is impaired in VMP1 and TMEM41B knockout cells. Analysis using transmission electron microscopy revealed that the formation of DMVs was substantially inhibited in cells lacking these autophagy proteins. Hence, by inhibiting VMP1 and TMEM41B expression, the virus is unable to hide from the immune sensors in DMVs, thus reducing its protection from the immune system [54,55]. Shneider et al., showed that TMEM41B participates in the transport of lipids to the membrane and that, together with VMP1, they are involved in the remodeling of the ER to form double-membrane vesicles (DMV) [38,52]. Scudellari compared the double-membrane spheres to bubbles being blown by the endoplasmic reticulum [20]. These DMVs may act as replication organelles (RO), which might provide a safe place for viral RNA to be replicated and translated, protecting it from innate immune sensors in the cell, similar to other β coronaviruses [56]. Therefore, these structures play a central role in infection, and, consequently, the loss of RO integrity due to the lack of VMP1 or TMEM41B could lead simultaneously to altered viral replication and enhanced antiviral signaling, as viral RNA is a very potent inducer of innate antiviral signaling [54,57]. However, the mechanism by which ER is transformed into these vesicles is still not fully elucidated.

SARS-CoV-2 mediates its replication through a dependent ATG5 pathway using specific DMVs that can be considered similar to autophagosomes. Mutations in the NSP6 protein with a positive influence on autophagosome production suggest a potential link with autophagy [45]. Thus, we hypothesize that some of these DMVs could be related to autophagy structures, and, more specifically, to autophagosomes. We are certain that, in the near future, it will be found that well-described autophagy markers colocalize with these DMVs in SARS-CoV-2-infected cells.

Another connecting pathway between the viral replication cycle and autophagy is represented by SNX27, one of the sorting nexin (SNX) family members, which down-regulates autophagy by increasing the level of mTORC1 signaling [58]. Figure 3, pathway 2 shows how SNX27 regulates the traffic of endosomal receptors towards recycling endosomes. Kim et al., found that mTORC1 acts as a signal integrator at the lysosome and can act as an inhibitor of later stages of autophagy, suppressing phosphorylation on UVRAG, which is a component of VPS 34 complex II. In this way, it avoids autophagosome and endosome maturation [59]. These events are relevant for the viral cycle, given that the virus enters the cell via directly fusing to the membranes in the cell surface pathway or via the endocytic pathway through endosome/lysosome-mediated endocytosis. Interestingly, it has recently been found that the ACE2 receptor possesses a type I PDZ binding motif (PBM) and can therefore interact with a PDZ domain-containing protein such as SNX27. A recent study has shown SNX27 to be critical for ACE2 cell surface regulation, and SNX27 prevents ACE2-bonded viral particles from entering the lysosome, down-regulating the endocytic viral entry pathway and therefore serving as a viral trafficking regulator [60].

Regarding autophagy machinery, a class III PI3-kinase that produces PI3P has a role in cellular trafficking and in the nucleation step in both canonical and non-canonical autophagy. In fact, inhibition of VPS34 kinase activity by VPS34-IN1, a well-known inhibitor for this kinase, reduced PI3P production and suppressed SARS-CoV-2 infection and replication in ex vivo human lung tissues [61] (Figure 3, pathway 5).

The features mentioned above provide consistent evidence that the autophagy machinery is actively involved in the viral entry and replication process of SARS-CoV-2 infection and therefore could be used as potential therapeutic targets to battle infection and prevent viral entry and replication at different steps. In Figure 3, we show the stage modeling of how SARS-CoV-2 is related to autophagy structures and molecules (Figure 3).

## 4. SARS-CoV-2 Infection and Its Effects on Autophagy

This section aims to describe the interplay between autophagy machinery proteins and newly described viral proteins and how this interaction affects viral replication and pathogenicity.

Although strong evidence points toward the SARS-CoV-2 virus having an inhibiting role in some stages of autophagy, paradoxically, it has been suggested that the virus enhances autophagy in other steps of this process. Li et al., explored the regulatory role of the SARS-CoV-2 spike protein in infected cells and attempted to elucidate the molecular mechanism of SARS-CoV-2-induced inflammation. They found that SARS-CoV-2 inhibits the PI3K/AKT/mTOR pathway by upregulating intracellular reactive oxygen species (ROS) levels, and, in this way, promotes the autophagic response. Subsequently, SARS-CoV-2-induced autophagy triggers inflammatory responses and apoptosis in infected cells [62].

Recent studies suggest that SARS-CoV-2 inhibits autophagy at different stages, limiting the autophagic flux to suppress viral clearance by selective autophagy, known as virophagy, a process mediated by autophagy receptors that recognize and sequester viral components inside autophagosomes. To avoid virus inactivation, SARS-CoV-2 uses the autophagy machinery for its benefit [14]. Autophagy also regulates adaptive immunity through antigen presentation. Gassen and colleagues found that SARS-CoV-2 modulates cellular metabolism and reduces autophagy; therefore, the induction of autophagy limits SARS-CoV-2 propagation [16].

Autophagy machinery proteins and viral proteins interact, leading SARS-CoV-2 to successfully survive and complete the replication cycle in the infected host cells.

Regarding SARS-CoV proteins, Mohamud and colleagues have shown that NSP3, one of the 16 nonstructural proteins, also known as papain-like protease (PLpro), can cleave the serine/threonine unc-51-like kinase (ULK1) and prevent the formation of the autophagy initiation complex in the absence of nutrients. In addition, PLpro showed deubiquitinase activity, which allows the virus to interrupt selective autophagy, preventing its proteins from being ubiquitinated [15]. Another recent study showed that different viral proteins targeted and inhibited autophagy to avoid viral clearance and to block the antiviral functions of autophagy [63]. SARS-CoV-2 uses autophagy to its benefit, hijacking the autophagy mechanism in the host cell to improve viral replication and to avoid the immune response and extracellular release. Viral proteins ORF3a and ORF7a were shown to cause the accumulation of autophagosomes [64]. Specifically, ORF3a interacts with autophagy-related protein UVRAG, suppressing autophagosome maturation and therefore the autophagy flux [53]. 

In two other studies, it was demonstrated that ORF3a interacts with VPS39, colocalizing with lysosomes. In this way, it impairs the binding of HOPS with RAB7, avoiding the regulation of the fusion of autophagosomes with lysosomes [65,66].

Another effect of viral protein ORF3a is its ability to promote lysosomal exocytosis, blocking autophagy flux and facilitating the lysosomal targeting of the BORC-ARL8b complex. Additionally, BORC-ARL8b is involved in lysosomal trafficking and modulates the exocytosis-related SNARE complex (VAMP7, STX4, and SNAP23). Following this pathway, the complex is oriented towards the plasma membrane area. This entire process is Ca 2+-dependent [67]. ORF7a generates a dysfunctional deacidified lysosome; therefore, autophagosomal degradation is interrupted and the virus can exit the host cell [64]. 

On the other hand, Koepke et al., used an mCherry-GFP-LC3B reporter system to show that lysosomal acidity, implicated in lysosomal degradation, was reduced in the presence of SARS-CoV-2. Regarding other viral protein effects, Nsp15 modulates autophagy regulation hypothetically by interfering with the mTOR pathway, in this way facilitating SARS-CoV-2 replication [64].

Non-structural protein NSP6 interacts with autophagy in different ways. It can join ER membranes, stimulating the rearrangement of its membranes and facilitating phagophore formation. This is another example of how SARS-CoV-2 uses the autophagy machinery to form DMVs to hide from the immune system and to replicate RNA. Moreover, it was found that viral protein NSP6 impairs autophagic flux, inhibiting autophagy at a late stage and impairing lysosomal acidification by targeting ATP6AP1, a vacuolar ATPase proton pump component. Consequently, the inflammasome is activated through NLRP3. To confirm that NSP6 elicits pyroptosis, Sun and colleagues experimentally overexpressed NSP6 and found that NLRP3/ASC Caspase-1-dependent activation released IL1β and IL18 in lung epithelial cells, thus being a crucial factor in viral pathogenicity [68].

It has been demonstrated that SARS-CoV-2 uses the mechanism of mitophagy, a well-known type of selective autophagy, as a strategy to regulate the host cell immune response. Hui and colleagues found that viral structural protein M joins with translation elongation factor (TUMF) M located in the mitochondrial external membrane and interacts with the LC3 II LIR domain; thus, the SARS-CoV-2 M protein breaks mitochondria networks by inducing mitophagy and then breaks downstream innate immunity signaling through inhibiting the type I IFN response [69]. Simultaneously, Li and colleagues demonstrated a similar effect of the ORF10 viral protein, which translocases to mitochondria and interacts with NIX—a protein very similar to the conforming protein of mitophagy receptor Nip3—and joins LC3 II. The activation of mitophagy leads to the degradation of mitochondrial antiviral signaling protein (MAVS), disrupting the activation of type I INF. This suppresses cellular pyroptosis and cytokine release, hijacking the immune response in favor of SARS-CoV-2 survival [70].

Additionally, SARS-CoV-2-infected cells are much less sensitive to lysis by cytotoxic T lymphocytes. This could be due to non-structural viral protein ORF8, which impairs antigen presentation with major histocompatibility complex I (MHCI) and leads MHCI to lysosomal degradation, mediated via autophagy. This mechanism also helps to evade the immune response [66]. ORF8 also mediates the escape from the immune system by degrading major histocompatibility complex (MHC) [71].

## 5. Pyroptosis, Autophagy, and SARS-CoV-2

SARS-CoV-2 is a cytopathic virus, which means that it produces cell and tissue death as part of its replication cycle [72]. In order to restrain the infection, infected cells can undergo a type of inflammatory programmed cell death called pyroptosis [73,74].

Pyroptosis produces an inflammatory response led by the secretion of interleukin 1 beta (IL1β) and interleukin 18 (IL18) and the recruitment of immune cells such as neutrophils, eosinophils, and macrophages, causing an excessive immune response and potentially massive multiorgan failure [75]. There are two different types of pyroptosis: canonical pyroptosis, which involves inflammasomes triggered by pathogen-associated molecular patterns (PAMPs) and damage-associated molecular patterns (DAMPs), and non-canonical pyroptosis, which is triggered by LPS and activates Caspase-1 [76]. Non-canonical pyroptosis results in Caspase-11 activation in mice, or Caspase-4 and Caspase-5 activation in humans, in response to lipopolysaccharide (LPS), a component of the Gram-negative bacterial cell wall, but does not cleave pro-IL-1β or pro-IL-18 [77,78]. This pathway results in cell swelling and lysis, preventing the replication of pathogens inside the infected cells.

Canonical pyroptosis is induced by the formation of inflammasomes, which are large cytosolic multiprotein complexes assembled in response to infection and cellular stress. Inflammasomes are crucial for the activation of inflammatory caspases and the subsequent processing and release of pro-inflammatory mediators, such as interleukin-1β (IL-1β) and IL-18 [79]. Inflammasomes are assembled by pattern recognition receptors (PRR) such as NOD-like receptor protein 3 (NLRP3), which are sensors of exogenous and endogenous “danger” signals from PAMPs and DAMPs. Hyperinflammation caused by unrestrained inflammasome activation is linked with multiple inflammatory diseases, including inflammatory bowel disease, Alzheimer’s disease, and multiple sclerosis. It has been shown that patients with SARS-CoV-2 infection have high levels of IL1β because of the imbalance of autophagy and inflammasome formation caused by the virus.

Autophagy responds to PRR from PAMPs and DAMPs to control homeostasis. Almost all PRR can induce autophagy, either directly or indirectly. Autophagy presents antigens to PRR and contributes to pathogen clearing. In addition, the induced autophagy forms a negative feedback regulation of PRR-mediated inflammation in a cell-/disease-specific manner to maintain homeostasis and prevent excessive inflammation. Understanding the interaction between PRR and autophagy in a specific disease can promote drug development for immunotherapy [80].

It has been proposed that autophagy modulates IL-1β production by means of inflammasome ubiquitination and the recruitment of p62 and LC3 [81,82]. Autophagy stimulates the clearance of inflammasomes, and inflammatory proteins related to pyroptosis [83]. In addition, some microorganisms, such as Streptococcus pneumoniae, increase ROS levels in the mitochondria of infected cells, causing mitochondrial damage and lower ATP production. In response to this damage, mitophagy is stimulated and consequently inactivates the NLRP3 inflammasome [84,85]. It was recently reported that protein E of SARS-CoV-2 activates the NLRP3 inflammasome [75,81,86]. Zhong et al., suggested that the parkin-dependent clearance of p62-bound mitochondria can reduce NLRP3 activation and IL-1β release in macrophages, providing another mechanism for the autophagy-mediated inhibition of inflammasome activation [87]. Later studies indicated that the regulation of inflammasome activation by autophagy can occur in multiple ways, through either the removal of endogenous inflammasome activators or removal of inflammasomes and their downstream cytokines directly. Saitoh et al., demonstrated that autophagy can negatively regulate pyroptosis. They found that when Atg16L1 is not present, it causes the hyperactivation of the inflammasome, with the excessive release of IL1β and IL18, and susceptibility to the development of intestinal inflammation was observed [88].

Autophagy also regulates non-canonical pyroptosis through the interaction of ATG8 with LPS and Caspase-11. The stimulation of autophagy increases ATG proteins and causes the inhibition of non-canonical pyroptosis [89].

Sun et al., reported pyroptosis induction in patients with COVID-19. They found that NSP6 overexpression in SARS-CoV-2 infection may affect lysosome acidification in lung epithelial cells by interacting with vacuolar ATPase, leading to the blockade of autophagic flux and inducing NLRP3 inflammasome activation and pyroptosis. When autophagy flux is pharmacologically restored, NSP6-induced pyroptosis is suppressed [90].

In contrast, there is some evidence that autophagy could promote pyroptosis. Dupont and colleagues have demonstrated that mice macrophages experience increased release of interleukins under starvation [91]. Finally, autophagy and pyroptosis could have a synergic effect in clearing microorganisms such as SARS-CoV-2. Xenophagy, the selective type of autophagy that degrades pathogens, may also be involved in SARS-CoV-2 infection. Moreover, when xenophagy is surpassed by pathogens, Caspase-1 might be cleaved and pyroptosis triggered.

## 6. Systemic Inflammatory Response Syndrome, SARS-CoV-2, and Autophagy

As described above, SARS-CoV-2 infection in the lung recruits immune cells and causes the overproduction of pro-inflammatory cytokines such as IL1β, IL2, IL10, and TNF. The imbalance in cytokine production causes a cytokine storm, leading to multiorgan damage, affecting the liver, heart, and kidneys [92].

Huang, in his prospective study, was the first to include an analysis of cytokine levels in severe and mild COVID-19, showing the presence of a cytokine storm analogous to that found for SARS-CoV-2 infection [93]. Excessive cytokine production could lead to multiorgan failure and worsen a patient’s prognosis. Moreover, the damage to epithelial and endothelial lung cells in SARS-CoV-2 infection, plus the cytokine storm, initiates the coagulation cascade via the tissular factor pathway [80]. Specifically, the excessive immune response characterized by an increase in IL1β, IL2, IL6, GSF, TNFα, and IFNɤ generates a local and systemic inflammatory response [94]. It is believed that these two inflammatory responses can cause hypercoagulability by increasing procoagulant molecules such as Von Willebrand factor, tissular factor, fibrinogen, and thrombin, leading to a reduction in blood flow and macro- and micro-thrombosis. In addition, there is an endotheliopathy caused by the vasoconstriction that also enhances the prothrombotic state. Masi and colleagues found that acute respiratory distress syndrome in COVID-19 was associated with procoagulants, specifically with plasminogen activator factor. Furthermore, they suggested that there is a potential role of endothelial dysfunction in the imbalance between procoagulant and anticoagulant agents. This procoagulant state can cause pulmonary embolism, acute respiratory distress syndrome (ARDS), and multiorgan failure, as is observed in patients with severe SARS-CoV-2 infection [95]. Additionally, the accumulation and infiltration of lymphocytes in the lungs leads to lymphopenia and neutrophilia, which leads to the formation of neutrophil extracellular traps (NETs). This process is called “NETosis”, which is a newly described type of programmed cell death that involves neutrophils as the key players in this mechanism, generating NETs by the extrusion of DNA, histones, and antimicrobial proteins, which are important for preventing pathogen infection. However, if NETs are in excess, a series of negative effects, such as autoimmune inflammation and tissue damage, could occur [96,97]. Interestingly, chronic metabolic diseases such as type 2 diabetes are also associated with high levels of NET production. Diabetes is characterized by inflammation, endothelial dysfunction, a risk of infection, and cardiovascular disease. NETosis is observed in patients with type 2 diabetes [98] and in SARS-CoV-2-infected individuals. Although it is still unclear whether this increased NETosis in type 2 diabetes patients is associated with the elevated incidence of thromboembolic events seen under SARS-CoV-2 infection, it has been postulated that NET overproduction may explain part of this increased risk. Similarly, non-diabetic obese patients have an increased incidence of thromboembolic events, as well as increased NETosis, and obesity is a risk factor for poor outcomes in COVID-19 patients. Moreover, it has been demonstrated that the increased level of angiotensin II seen in patients with hypertension could trigger NETosis, increasing the cardiovascular risk in these patients [99,100].

It has been found that the presence of plasma NETs correlates with high sequential organ failure assessment (SOFA) scores in COVID-19. Interestingly, there is an increase in plasma NETs in patients with SARS-CoV-2 and this correlates with the severity of the clinical presentation [101]. NETs mediate harmful effects caused by neutrophils, leading to unfavorable coagulopathy and immune thrombosis, and are a major element of micro- and macrovascular thrombi [102,103]. There is an intricate and complex relationship among SARS-CoV-2 infection, NETs, and cytokine storms, which is not entirely understood at present. When SARS-CoV-2 enters the cells, pyroptosis is activated, causing an increase in cytokines released by endothelial and epithelial lung cells, which recruit and activate neutrophils, triggering NETosis [104], which increases the severe effects of SARS-CoV-2 infection [105].

It is important to highlight that autophagy participates in NET formation and regulation. An increased concentration of IL8 due to SARS-CoV-2 infection induces autophagy and then autophagy triggers NETosis in neutrophils [106]. In addition, secretory autophagy plays an important role in the release of IL1β, enhancing the activation and recruitment of neutrophils [91]. Nevertheless, autophagy regulates the externalization of membrane-bound and cytosolic proteins, modulating the NET vacuolation process. Autophagy may also have a role in limiting the respiratory burst, preventing cytoskeletal dynamics and chromatin decondensation, and generating histone citrullination [68]. Additionally, Remijsen et al., detected defective intracellular chromatin decondensation when they suppressed neutrophil autophagy, elucidating that autophagy is essential for the first stage of NETosis (DNA decondensation), rather than the cell killing process itself [107].

It has been suggested that the PI3K-AKT-mTOR axis links autophagy with NET formation and has a significant impact on both processes. The serine/threonine kinase mTOR controls cellular stress, proliferation, and autophagy. On the other hand, mTOR down-regulates autophagy when activated by dephosphorylation. However, Rapamycin and WYE-354 are autophagy inducers that inhibit mTOR and enhance NET formation in human neutrophils via autophagy downstream of formyl peptide receptor (FPR) signaling [68].

In contrast with the previous evidence, it has also been found that autophagy modulates NETosis to prevent an excessive immune response. For instance, as was discussed above, the viral proteins M and ORF10 could interfere in the development of NET formation by stimulating mitophagy and inhibiting the release of IL1β. Kim and colleagues have discovered an increase in autophagy activity and NET formation in hemodialysis patients. Interestingly, when autophagy was blocked, neutrophil activity and NET release rose significantly, demonstrating that autophagy may also play a role in limiting excess NET production [59].

To summarize, NETosis is a beneficial immune process triggered by inflammation and infection to protect the host organism and cells against pathogens such as SARS-CoV-2. However, the overproduction of NETs could cause an immune thrombotic state that exacerbates the systemic inflammation caused by severe COVID-19. Considering that autophagy can prevent or limit NET production, the activation of autophagy may play an essential role in modulating systemic inflammation. Taken together, it is important to highlight the relationship between the immune system, inflammation, and autophagy, to treat and improve the outcomes of patients with SARS-CoV-2 infection.

The interplay between local and systemic inflammatory responses, SARS-CoV-2, and autophagy is summarized in Figure 4.

## 7. Post-COVID-19 Syndrome

Post-COVID-19 syndrome is described as symptoms that are persistent for many weeks or months after the acute disease has resolved [108,109,110]. The World Health Organization (WHO) defined it in October 2021 as “Illness that occurs in people who have a history of probable or confirmed SARS-CoV-2 infection; usually within three months from the onset of COVID-19, with symptoms and effects that last for at least two months” [67].

The pathogenetic mechanism of post-COVID-19 pulmonary fibrosis is currently a topic of intense research interest but is still largely unexplored (recently reviewed by [111]). Although they are still poorly understood, the immunological dysregulations are probably associated with post-COVID-19 syndrome. In particular, patients with a prolonged symptom duration maintained antigen-specific T-cell response magnitudes to the virus in CD4+ and increased T follicular helper cell (Tfh) populations throughout late convalescence, while those experiencing a full recovery demonstrated a decline in these cellular populations [112,113,114,115]. The symptoms can vary among people experiencing post-COVID-19 syndrome, but there are general symptoms that are the most prevalent. The five most common symptoms are fatigue (58%), headache (44%), attention disorder (27%), hair loss (25%), and dyspnea (24%) [116].

Respiratory and cardiovascular symptoms include cough, dyspnea, and chest pain, among others, and neuropsychiatric symptoms range from difficulty thinking, sleep problems, and changes in smell or taste, to symptoms of clinical depression. There are certain specific patient populations that present a higher risk of developing this condition. These include patients with underlying autoimmune diseases, patients who have had severe COVID-19, people who needed intensive care, and unvaccinated patients, among others [116,117,118]. The pathophysiological mechanisms that underlie post-COVID-19 syndrome include endothelial damage, direct viral toxicity, a pro-inflammatory and prothrombotic state, and immune system dysregulation. Apparently, severe COVID-19 infection survivors have an increased risk of presenting bacterial, fungal, and viral infections thereafter [111,112,114,115].

As described above, autophagy regulates the immune response and NETosis in SARS-CoV-2 infection, and its dysregulation by the virus itself can worsen the clinical outcomes of patients. It is believed that long COVID syndrome is more common in patients who have had severe symptoms and who have required long-term hospitalization because they have had a prolonged inflammatory response [117]. Taken together, it could be hypothesized that autophagic dysregulation could increase the risk of developing long COVID or post-COVID-19 syndrome. Thus, maintaining autophagy within normal functioning could aid in preventing long-term sequelae by controlling the inflammatory and immune response, and therefore diminishing post-COVID syndrome. This needs to be elucidated in future investigations.

## 8. Conclusions

Autophagy would be involved in the entry, as well as transcription and translation, of viral particles in a host cell [119]. Additionally, autophagy could be involved in the systemic inflammatory response and post-COVID-19 syndrome. Selective autophagy, especially mitophagy, was reported to be induced by SARS-CoV-2 proteins, modulating the inflammatory response. Secretory autophagy may also be involved in the development of the thrombotic immune-inflammatory syndrome seen in a significant number of COVID-19 patients that leads to severe illness and even death.

However, autophagy does not only interact with SARS-CoV-2 infection by participating in its viral replication cycle. Studies suggest that, surprisingly, reciprocal dysregulation of autophagy by the viral infection itself could be one of the mechanisms of viral survival and tissue damage, given the antimicrobial functions of autophagy, its ability to aid with viral clearance through xenophagy, and its immunological role in battling infection and regulating excessive inflammation [120]. Autophagy’s role as a balancer of the beneficial and detrimental effects of immunity and inflammation becomes disrupted by viral effects on autophagy’s complex machinery [120,121]. Dysregulation of autophagy could imply the disinhibition of pyroptosis, excess NETosis, and other molecular processes, stimulating the release of pro-inflammatory cytokines and interleukins that favor an exaggerated inflammatory response overall, leading to a thrombotic immunoinflammatory state that correlates with more severe clinical illness in COVID-19. In fact, one of the described cargoes of unconventional autophagy-associated secretion pathways is the export of cytosolic protein IL1β, a pro-inflammatory cytokine, which has a central role in inducing pro-inflammatory signaling [122]. A possible link between this process and the cytokine storm that characterizes the immunoinflammatory state seen in patients with severe COVID-19 could be further explored.

Consequently, identifying different autophagic biomarkers could help to correlate with the severity of illness, and thus serve as a biological marker for the prognosis of the disease. Many viruses with induced direct and indirect mechanisms explaining most of the short-term complications of the disease have correlations with alterations in autophagy. Long-term post-COVID-19 syndrome may also be related to dysfunctional autophagy. Associations between interindividual markers of short- and long-term prognosis and dysfunctional autophagy offer many gaps for further investigation. More research is needed to clarify the involvement of these abnormalities in disease infection and clinical evolution.

## Figures and Tables

**Figure 1 ijms-24-04928-f001:**
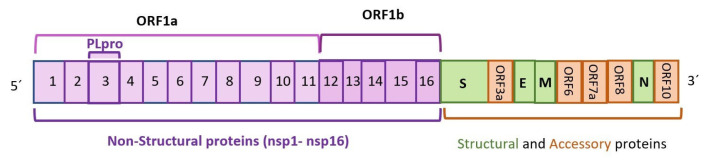
Genome of SARS-CoV-2 virus showing ORF1a and ORF1b open reading frames.

**Figure 2 ijms-24-04928-f002:**
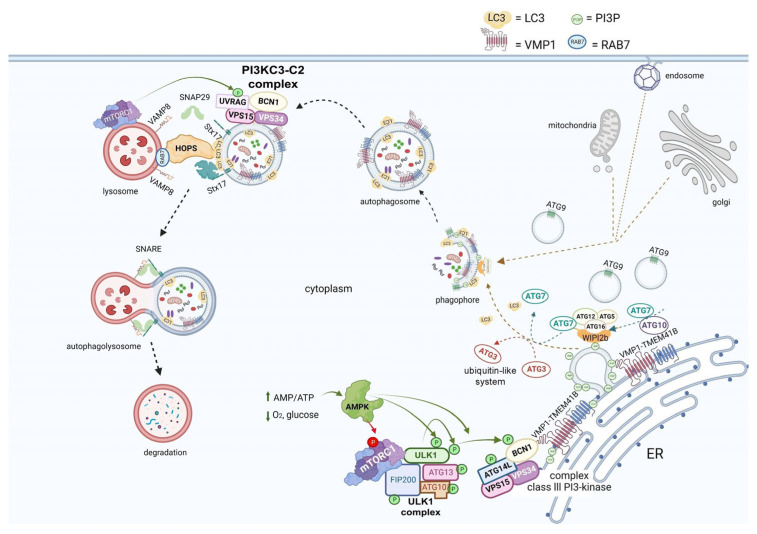
**Autophagy overview diagram flow**. Autophagy can be initiated by a lack of nutrients, implying amino acid, glucose, or O2 depletion, or an AMP/ATP increase, as a result. When AMPK is activated, it inhibits the mTORC1 complex and activates the ULK1 complex through phosphorylation. Autophagosome biogenesis is mediated by ULK1 activation and translocation to ER. Thus, the class III PI3-kinase complex is activated by the phosphorylation of BECN1 and ATG14L; this process leads to a PI3P production increase through catalytic subunit VPS34 activation; transmembrane protein VMP1 interacts with BECN1 and WIPI2b protein is recruited. WIPI recruits ATG16L-ATG5-ATG12, a ubiquitin-like system, on the isolation membrane, mediating LC3 lipidation on the membrane. LC3 plays a crucial role in the whole autophagy process. The transmembrane protein ATG9 is implied in vesicular trafficking towards the phagosome maturation area. Once the autophagosome is mature, the second complex, PI3KC3-C2, replaces ATG14L by UVRAG, which is activated by mTORC1 and participates in autophagosome–lysosome fusion. This fusion is regulated by the HOPS complex. STX17 interacts with lysosomal VAMP8 through SNAP29, constituting the SNAREs, forming the autolysosome structure. The complexes highlighted in this figure are as follows: complex ULK1 (ULK1-ATG13-FIP200-ATG10); complex class III PI3-kinase (BECN1-ATG14L-VPS34-VPS15); complex PI3KC3-C2 (BECN1-UVRAG-VPS34-VPS15). Figure was created with BioRender.com.

**Figure 3 ijms-24-04928-f003:**
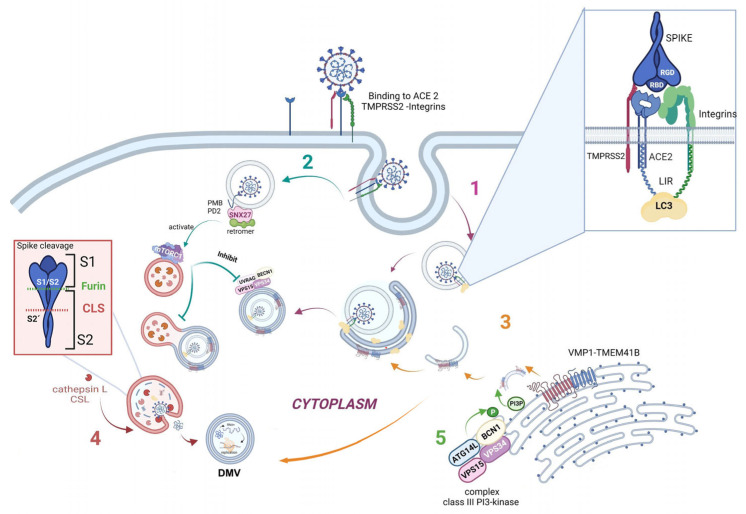
**Schematic modeling of the crosstalk between autophagy and SARS-CoV-2 replication pathways**. 1—Autophagy may help virus endosomal entry through LC3. 2—Autophagy may be down-regulated by the activation of SNX27 present in the endosomes. 3—VMP1 and TMEM41B trigger autophagosome formation to aid viral replication. 4—The protease Cathepsin L allows viral RNA release to the cytosol, before viral replication into DMVs. 5—The complex class III PI3-kinase through VPS34 is necessary for SARS-CoV-2 infection and replication. Figure was created with BioRender.com.

**Figure 4 ijms-24-04928-f004:**
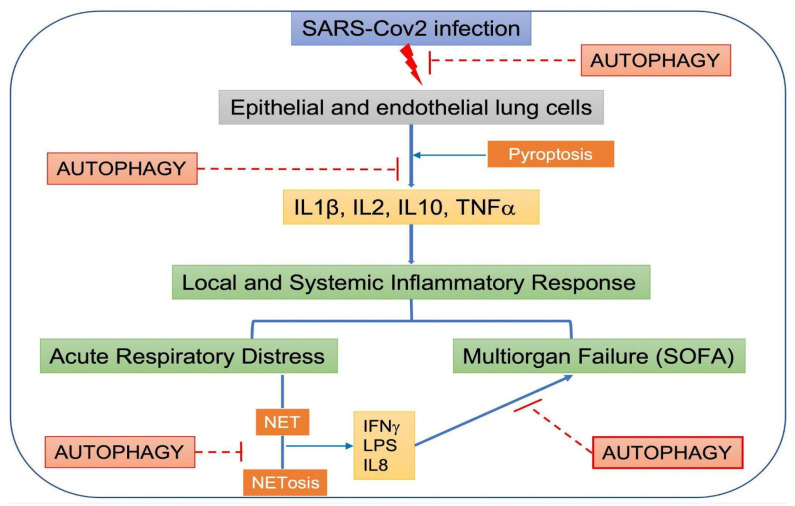
**Schematic model of the crosstalk among systemic inflammatory response syndrome, SARS-CoV-2, and autophagy**. There is evidence that autophagy could participate in the sequential systemic inflammatory events triggered by SARS-CoV-2 infection that eventually would lead to multiorgan failure. Here, we outline the steps in which autophagy can function as a protective cellular response to COVID-19.

## Data Availability

Not applicable.

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
