# Peer review of "Autophagy in Inflammatory Response against SARS-CoV-2"

_ijms, 2023, doi:10.3390/ijms24054928_

Round 1
Reviewer 1 Report
Since COVID-19 pandemic outbroken, people started to focus on how SARS-CoV-2 virus invaded the cell and how cell defense themselves from the virus. In the field of Autophagy, scientists were also have their own opinions about the importance of Autophagy both in the virus infection and cell defense sides. This review summarised a lot of the knowledge of the relationship between autophagy and SARS-CoV-2 virus infection. Basically, the review is well organized and comprehensively described. I am sure scientists in this field will appreciate this work. However, my main concern is, what are the advantages and/or differences of your review compare to the other recent reviews of the same topic, such as Di Chen et al, 2022 (https://doi.org/10.1016/j.cophys.2022.100596), Qinqin Sun et al, 2022 (https://doi.org/10.1080/15548627.2022.2116677), Weifeng He et al, 2022 (https://doi.org/10.7150/ijbs.72544), Anna GorÄ…cy et al, 2022 (https://doi.org/10.3390/v14051092), etc.?
Additionally, a lot of important states (sentences or even paragraphs) lack citation. I mentioned some of them in the following point-to-point comments, but there are more. Please check about your citation carefully. This is very important and critical for a Review type paper.
Other than that, I have the following comments or suggestions:
1. The 1st paragraph, Please place the citations in front of the period mark (.).
2. Line 37, please replace the period mark (.) with semicolon (;), because the later "accessory proteins" and "structural proteins" are all belong to the proteins encoded by the "11 genes with 14 reading frames".
3. Line 37, how many accessory proteins?
4. Line 43-44, please cite for the function of ORF7a.
5. Line 60, please name the step 2, e.g. "2. Virus entry: the virus has two entry pathways...".
6. Line 72, the citation "Khade et al, 2021; Shereen et al. 2020; Iqbal et al. 2020" should go to Line 70.
7. In my opinion, the "Overview of Autophagy" should be combined into the introduction. Moreover, you should mention the works about the relationship between autophagy and virology, e.g. virus utilize autophagy to infect cell (not limited to human and SARS-CoV-2) (sample citation: PMID: 36471479), and cell defend virus infection via autophagy (sample citation: PMID: 17921696). Minimum, if you want to keep this section, you still need to briefly introduce autophagy in the introduction part, and the relationship between autophagy and virus-cell battling.
8. Line 86-89, please cite for this paragraph.
9. Line 95-96, please cite for microautophagy.
10. Figure 2, is the number 2 labeled at a wrong place? Based on your description, it should be next to the enlarged Spike protein structure depiction rather than at the virus entry part which should be belonging to Number 1 step.
11. Line 218, what is the function of the cleavage?
12. Line 221, how did the virus use VMP1 and TMEM41B to evade the immune system?
13. Line 230-234, is there any mechanism elaborated about the "ER bubbling"?
14. Line 233, you claimed the virus replicates its RNAs in the DMV, but in Figure 2, it seems that the virus RNA released to cytosol before RNA replication. Please clarify.
14. Line 236-237, is this sentence reported by literature or just your opinion? If there are supporting literatures, please quote and cite; if it is your opinion, please further explain why you think so. For example, are there any autophagy related marker on the DMVs?
15. Figure 2, at the number 4 position, it seems that autophagy also helps the virus uncoating. Are there any literatures supporting?
16. Line 284, Which ORF does PLpro peptide belong to? I think it is a better idea to show a scheme to describe the ORFs of the SARS-CoV-2 virus genome.
17. Line 287-288, maybe this sentence should be combined to the last paragraph, since they are both talking about PLpro.
18. Line 289, "Another recent study" which one? Please cite.
19. Line 289-297, this part lacks citations.
20. Line 373-375, Huang paper only reported the hyperinflammation caused by COVID-19, but they did not mention about autophagy. Inflammation is not always related to autophagy. So I think there is no adequate evidence to claim that SARS-CoV-2 induces inflammation via autophagy at this stage. Perhaps you can move this paragraph after later evidences (or even in section 5).
21. Line 404-408, is there any reports talking about SARS-CoV-2 regulates non-canonical pyroptosis through autophagy?
22. Line 448, please replace "COVID" with "SARS-CoV-2".
23. Line 443-451, this is an interesting point. Many prevalence of cardiovascular disease can increase the risk of COVID-19 symptoms. So is there any evidences that hypertension also have an increased NETosis?
24. Line 464-466, please cite this part.
25. Line 524, please verify whether the symptoms of "fever, fatigue malaise or systemic specific symptoms" are COVID-19 syndromes or post-COVID-19 syndromes.
26. Line 550, I don't think you have ever mentioned about autophagy being involved in viral particle releasing.
Author Response
We thank you very much for the comments, which allow us to significantly improve our manuscript. We have addressed the comments in the revised version of our manuscript and all the corrections are highlighted in yellow.
Comments and Suggestions for Authors: Since COVID-19 pandemic broke out, people started to focus on how SARS-CoV-2 virus invaded the cell and how cells defense themselves from the virus. In the field of Autophagy, scientists also have their own opinions about the importance of Autophagy both in the virus infection and cell defense sides. This review summarised a lot of the knowledge of the relationship between autophagy and SARS-CoV-2 virus infection. Basically, the review is well organized and comprehensively described. I am sure scientists in this field will appreciate this work. However, my main concern is, what are the advantages and/or differences of your review compare to the other recent reviews of the same topic, such as Di Chen et al, 2022 (https://doi.org/10.1016/j.cophys.2022.100596), Qinqin Sun et al, 2022 (https://doi.org/10.1080/15548627.2022.2116677), Weifeng He et al, 2022 (https://doi.org/10.7150/ijbs.72544), Anna GorÄ…cy et al, 2022 (https://doi.org/10.3390/v14051092), etc.?
We understand that since COVID-19 pandemic broke out, people started to focus on how SARS-CoV-2 virus invaded the cell and how cells defend themselves from the virus and several excellent reviews were published during 2022 covering the interplay between viral proteins and autophagy and their pathogenic effects; the membrane proteins targeted by the virus; and the mechanisms of vesicles and organelles related to autophagy during infection of coronaviruses. In the case of our manuscript, it focuses on the crosstalk of canonical and noncanonical autophagy molecular pathways on the cellular and systemic inflammatory response against SARS-Cov-2. We are including complex mechanisms of cell death, such as pyroptosis and NETosis, and briefly describing key concepts on autophagy and its relationship among these mechanisms in the context of SARS-Cov-2 infection. We also discuss the relationship between autophagy and the systemic inflammatory response in the severe COVID-19, also noting the reciprocal effect of viral infection in autophagic pathways and their clinical aspects.
Additionally, a lot of important states (sentences or even paragraphs) lack citation. I mentioned some of them in the following point-to-point comments, but there are more. Please check about your citation carefully. This is very important and critical for a Review type paper.
We deeply appreciate this suggestion and comments. In the revised version of our manuscript, we have checked our citations carefully, we added citations and we addressed all the point-to-point comments as follows:
- The 1st paragraph, please place the citations in front of the period mark (.).
All citations have been placed in front of the period mark in the revised version of the manuscript.
- Line 37, please replace the period mark (.) with semicolon (;), because the later "accessory proteins" and "structural proteins" are all belong to the proteins encoded by the "11 genes with 14 reading frames".
Thank you very much for this comment. To clarify, we corrected and changed the paragraph in lines 40 to 49 as follows:
“Its genome consists of 11 genes with 11 reading frames that produce 16 non-structural proteins (NSP1 to NSP16) and 4 structural proteins including the fusion trimeric spike (S), the envelope protein (E), the nucleocapsid protein (N), and the membrane glycoprotein (M) (Yoshimoto et al., 2020). These reading frames generate eight accessory proteins: ORF3a, ORF3b, ORF6, ORF7a, ORF7b, ORF8a, ORF8b, and ORF9b (Yoshimoto et al., 2020; Lv et al., 2022). NSPs are key for viral RNA replication and immune avoidance, and the accessory proteins play diverse roles, aiding in viral infection, replication, and transmission (Thomas et al., 2021).”
- Line 37, how many accessory proteins?
This comment was addressed in the revised manuscript, along with comment 4 described above, please see lines 44-45.
“…generate eight accessory proteins: ORF3a, ORF3b, ORF6, ORF7a, ORF7b, ORF8a, ORF8b, and ORF9b (Yoshimoto et al., 2020; Lv et al., 2022).”
- Line 43-44, please cite for the function of ORF7a.
A cite for the function of ORF7a (Koepke et al., 2021) was added in line 322 of the revised version of the manuscript:
“Koepke, L.; Hirschenberger, M.; Hayn, M.; Kirchhoff, F.; Sparrer, K.M. Manipulation of autophagy by SARS-CoV-2 proteins. Autophagy., 2021 ;17(9):2659-2661. doi: 10.1080/15548627.2021.1953847. “
- Line 60, please name step 2, e.g., "2. Virus entry: the virus has two entry pathways...".
Thank you. This correction was made and now it can be found in line 78.
- Line 72, the citation "Khade et al, 2021; Shereen et al. 2020; Iqbal et al. 2020" should go to Line 70.
This correction was addressed, and the citation is now in line 88-89.
- In my opinion, the "Overview of Autophagy" should be combined into the introduction. Moreover, you should mention the works about the relationship between autophagy and virology, e.g. viruses utilize autophagy to infect cells (not limited to humans and SARS-CoV-2) (sample citation: PMID: 36471479), and cells defend virus infection via autophagy (sample citation: PMID: 17921696). Minimum, if you want to keep this section, you still need to briefly introduce autophagy in the introduction part, and the relationship between autophagy and virus-cell battling.
Thank you very much for these comments and suggestions. In the revised version of our manuscript, we transferred a paragraph from the section “Overview of Autophagy” to the “Introduction”, adding a new paragraph to the introduction, describing briefly canonical and non-canonical autophagy, and mentioning papers on the relationship between autophagy and virus-cell battling (lines 54 to 64):
“Autophagy is a catabolic process that sequesters damaged cell organelles, proteins, and external invading microbes and delivers them to the lysosomes for degradation (Shojaei et al., 2020; Grasso et al., 2018; Yu et al., 2022; Klionsky et al., 2007). It is a fundamental and evolutionarily conserved eukaryotic cellular process that has multiple effects on cell survival, homeostasis, and immunity (Glick et al., 2010). Noncanonical autophagy describes other processes that use the autophagy molecular machinery such as phagocytosis, inflammatory signaling, and secretion (Debnath et al., 2022). Autophagy plays an essential role in promoting RNA virus replication by inhibiting innate anti-virus immune responses (Takeshita et al., 2008), or promoting infectivity by the autophagy-related secretion of vesicles loaded with virus (Dahmane et al., 2022; Morita et al., 2018; Mohamud et al., 2021; Gassen et al., 2021)."
- Line 86-89, please cite for this paragraph.
The citation for this paragraph was added in line 56.
Shojaei et al., 2020; Grasso et al., 2018; Yu, G. et al., 2022; Klionsky et al., 2007).
and in line 58 (Glick D et al., 2010).
Grasso, D.; Renna, F.J.; Vaccaro, M.I. Initial Steps in Mammalian Autophagosome Biogenesis. Front Cell Dev Biol. 2018 ;6:146. doi: 10.3389/fcell.2018.00146.
Klionsky, D. Autophagy: from phenomenology to molecular understanding in less than a decade. Nat Rev Mol Cell Biol 2007, 8, 931–937. https://doi.org/10.1038/nrm2245.
Glick D, Barth S, Macleod KF. Autophagy: cellular and molecular mechanisms. J Pathol. 2010 May;221(1):3-12. doi: 10.1002/path.2697. PMID: 20225336; PMCID: PMC2990190.
Shojaei, S.; Suresh, M.; Klionsky, D.J.; Labouta, H.I., Ghavami, S. Autophagy and SARS-CoV-2 infection: Apossible smart targeting of the autophagy pathway. Virulence. 2020;11(1):805-810. doi: 10.1080/21505594.2020.1780088.
Yu, G.; Klionsky, D.J. Life and Death Decisions-The Many Faces of Autophagy in Cell Survival and Cell Death. Biomolecules. 2022 Jun 21;12(7):866. doi: 10.3390/biom12070866. PMID: 35883421; PMCID: PMC9313301.
- Line 95-96, please cite for microautophagy.
The citation was added (Li, W.W et al., 2012) in the line 104 of the revised version of the manuscript:
“Li, W.W.; Li, J.; Bao, J.K. Microautophagy: lesser-known self-eating. Cell Mol Life Sci. 2012;69(7):1125-1136. doi:10.1007/s00018-011-0865-5.”
- Figure 2, is the number 2 labeled at a wrong place? Based on your description, it should be next to the enlarged Spike protein structure depiction rather than at the virus entry part which should be belonging to Number 1 step.
Thank you very much for this comment. As a clarification, we changed the legend in Figure 2 (Figure 3 in the revised manuscript) explaining the reason of the location of Number 2 as follows:
“Figure 3: Schematic modeling of the cross talking between Autophagy and SARS-CoV-2 replication pathways. 1-Autophagy may help virus endosomal entry through LC3. 2-Autophagy may be downregulated by the activation of SNX27 present in the endosomes. 3-VMP1 and TMEM41B trigger autophagosome formation to aid viral replication. 4-The protease Cathepsin L allows viral RNA release to the cytosol, before viral replication into DMVs. 5- The complex class III PI3-kinase through VPS34 is necessary for SARS-CoV-2 infection and replication.”
- Line 218, what is the function of the cleavage?
This item was addressed in lines 199 – 207, modifying the following paragraph:
“As previously described, the virus enters the cell via the endosomal pathway. The spike protein is composed of two subunits: one binding subunit (S1) and one fusion subunit (S2). First, the S1 subunit binds to the host cell receptor ACE2. This union induces conformational changes in protein S. Second, the S2 subunit is activated. This activation occurs through two consecutive cleavage steps: a cleavage between the S1 and S2 domains of protein S by Furin is produced, and then undergoes further cleavage at the S2´ site that promotes the unmasking and activation of the fusion peptide (Gomes et al., 2020; Hoffmann et al., 2020). To activate the spike protein fusion potential, a second cleavage performed by the host’s proteases is required.”
- Line 221, how did the virus use VMP1 and TMEM41B to evade the immune system?
Thank you for these comments. In the revised version of the manuscript, we have rewritten this paragraph to clarify the role of these protein in lines 219 to lines 235 and added the corresponding citations:
“Once the virus has entered the cell, it uses the autophagy machinery for its own benefit through viral proteins NSPs. Recently, as shown in Figure 3, pathway 3, transmembrane proteins related to autophagy VMP1 and TEMEM41B have been reported as critical host factors at the early stages of viral infection (Schneider et al., 2021). VMP1 and TMEM41B both contribute to different stages of DMV formation. Ji et al. have revealed that DMV biogenesis is impaired in VMP1 and TMEM41B knockout cells. Analysis using transmission electron microscopy revealed that the formation of DMVs was substantially inhibited in cells lacking these autophagy proteins. Hence, by inhibiting VMP1 and TMEM41B expression, the virus is unable to hide from the immune sensors in DMVs, thus reducing its protection from the immune system (Ji et al., 2022; Trimarco et al., 2021). Shneider et al. showed that TMEM41B participates in the transport of lipids to the membrane and that together with VMP1, they are involved in the remodeling of the ER to form double-membrane vesicles (DMV) (Hama et al., 2022; Schneider et al., 2021).”
The following references were added:
“Hama, Y.; Morishita, H.; Mizushima, N. Regulation of ER-derived membrane dynamics by the DedA domain-containing proteins VMP1 and TMEM41B. EMBO Rep. 2022 Feb 3;23(2):e53894. doi: 10.15252/embr.202153894”
“Trimarco, J.D.; Heaton, B.E.; Chaparian, R.R.; Burke, K.N.; Binder, R.A.; Gray, G.C. et al. TMEM41B is a host factor required for the replication of diverse coronaviruses including SARS-CoV-2. PLoS Pathog. 2021 May 27;17(5):e1009599. doi: 10.1371/journal.ppat.1009599. PMID: 34043740; PMCID: PMC8189496.”
“Scutigliani, E.M.; Kikkert, M. Interaction of the innate immune system with positive-strand RNA virus replication organelles. Cytokine Growth Factor Rev. 2017 Oct;37:17-27. doi: 10.1016/j.cytogfr.2017.05.007. Epub 2017 Jun 27. PMID: 28709747; PMCID: PMC7108334.“
“Snijder, E.J.; Limpens, R.W.A.L.; de Wilde, A.H.; de Jong, A.W.M.; Zevenhoven-Dobbe, J.C.; Maier, H.J. et al. A unifying structural and functional model of the coronavirus replication organelle: Tracking down RNA synthesis. PLoS Biol. 2020 Jun 8;18(6):e3000715. doi: 10.1371/journal.pbio.3000715. PMID: 32511245; PMCID: PMC7302735.”
“Ji, M.; Li, M.; Sun, L.; Zhao, H.; Li, Y.; Zhou, L. et al. VMP1 and TMEM41B are essential for DMV formation during β-coronavirus infection. J Cell Biol. 2022 Jun 6;221(6):e202112081. doi: 10.1083/jcb.202112081. Epub 2022 May 10. PMID: 35536318; PMCID: PMC9097365.”
- Line 230-234, is there any mechanism elaborated about the "ER bubbling"?
Thank you for this comment. We change this expression in the revised manuscript, and we added the following paragraph to address your comment in line 228 to 240.
“Shneider et al. showed that TMEM41B participates in the transport of lipids to the membrane and that together with VMP1, they are involved in the remodeling of the ER to form double-membrane vesicles (DMV) (Hama et al., 2022; Schneider et al., 2021). Scudellari compared the double-membrane spheres to bubbles being blown by the endoplasmic reticulum (Scudellari et al., 2021). These DMVs may act as replication organelles that might provide a safe place for viral RNA to be replicated and translated, protecting it from innate immune sensors in the cell, similar to other β coronaviruses (Snjider et al., 2020). Therefore, these structures play a central role in infection and, consequently, the loss of RO integrity due to the lack of VMP1 or TMEM41B could lead simultaneously to altered viral replication and enhanced antiviral signaling, as viral RNA is a very potent inducer of innate antiviral signaling (Ji et al., 2022; Scutigliani et al., 2017). However, the mechanism by which ER is transformed into these vesicles is still not fully elucidated.”
14 A. In Line 233, you claimed the virus replicates its RNAs in the DMV, but in Figure 2, it seems that the virus RNA is released to cytosol before RNA replication. Please clarify.
Thank you very much for this observation. Regarding this, the Figure 2 was modified according to your comments. Specially, pathway 4 was clarified in this figure. It was also corrected in the legend of the revised Figure 2, which is Figure 3 in the revised version of the manuscript.
14 B. Line 236-237, is this sentence reported by literature or just your opinion? If there are supporting literatures, please quote and cite; if it is your opinion, please further explain why you think so. For example, are there any autophagy-related marker on the DMVs?
We apologize for this mistake. We modified this sentence and added the supporting citation. The modified sentence in the revised version of the manuscript is in lines 241- 247 as follows:
“SARS-CoV-2 mediates its replication through a dependent ATG5 pathway using specific DMVs that can be considered similar to autophagosomes. Mutations in the NSP6 protein with a positive influence on autophagosome production suggest a potential link with autophagy (Sargazi et al., 2021). Thus, we hypothesize that some of these DMVs could be related to autophagy structures, and, more specifically, to autophagosomes. We are certain that in the near future it will be found that well described autophagy markers colocalize with these DMVs in SARS-CoV-2-infected cells.”
Reference (Sargazi et al., 2021) was added:
“Sargazi S, Sheervalilou R, Rokni M, Shirvaliloo M, Shahraki O, Rezaei N. The role of autophagy in controlling SARS-CoV-2 infection: An overview on virophagy-mediated molecular drug targets. Cell Biol Int. 2021 Aug;45(8):1599-1612. doi: 10.1002/cbin.11609. Epub 2021 Apr 23. PMID: 33818861; PMCID: PMC8251464.”
- Figure 2, at the number 4 position, it seems that autophagy also helps the virus uncoating. Are there any literatures supporting this?
We appreciate your comment, which significantly improves our manuscript. We have revised the paragraph accordingly and have included the corresponding citation. It is in lines 206-213 in the revised manuscript as follows.
“To activate the spike protein fusion potential, a second cleavage performed by the host’s proteases is required. The cleavage can occur at different stages of the virus infection cycle by different host proteases, such as Furin (convertase), TMPRSS2 (cell surface protease), and Cathepsin L (lysosomal protease) (Peacock et al., 2021; Bestle et al., 2020). Cathepsin L, which links the virus cycle to the autophagy process, acts at low pH, degrades cargo, and maintains autolysosome homeostasis and autophagic flux (Xu et al., 2021). By cleaving the spike protein at S2´, it mediates virus membrane and autolysosome fusion, thus facilitating the release of viral RNA into the host cell.”
The following references were added:
“Xu, T.; Nicolson, S.; Sandow, J.J.; Dayan, S.; Jiang, X.; Manning, J.A. et al. Cp1/cathepsin L is required for autolysosomal clearance in Drosophila. Autophagy, 2021 ;17(10):2734-2749. doi: 10.1080/15548627.2020.1838105.”
“Bestle, D.; Heindl, M.R.; Limburg, H. Van Lam van, T.; Pilgram, O.; Moulton, H. et al. TMPRSS2 and furin are both essential for proteolytic activation of SARS-CoV-2 in human airway cells. Life Sci Alliance, 2020, 3(9):e202000786. doi: 10.26508/lsa.202000786.”
“Peacock, T.P.; Goldhill, D.H.; Zhou, J. et al. The furin cleavage site in the SARS-CoV-2 spike protein is required for transmission in ferrets. Nat Microbiol ,2021; 6, 899–909. doi: 10.1038/s41564-021-00908-w.”
- Line 284, Which ORF does PLpro peptide belong to? I think it is a better idea to show a scheme to describe the ORFs of the SARS-CoV-2 virus genome.
Thank you for these comments.
In the revised manuscript we explain that PLpro is a viral nonstructural protein also known as NSP3, and we discuss its role in cleaving ubiquitinylated proteins in the following paragraph (lines 312-317).
“Regarding SARS-CoV proteins, Mohamud and colleagues have shown that NSP3, one of the 16 nonstructural proteins, also known as papain-like protease (PLpro), can cleave the serine/threonine unc-51-like kinase (ULK1) and prevent the formation of the autophagy initiation complex in the absence of nutrients. In addition, PLpro showed deubiquitinase activity, which allows the virus to interrupt selective autophagy, preventing its proteins from being ubiquitinated (Mohamud et al., 2021).”
Additionally, we provide a diagram to illustrate the ORFs of the SARS-CoV-2 virus genome. It was incorporated as Figure 1 in the Introduction section. The number of the subsequent figures was changed accordingly.
Figure 1: Genome of SARS-CoV-2 virus showing ORF1a and ORF1b open reading frame.
- Line 287-288, maybe this sentence should be combined with the last paragraph, since they are both talking about PLpro.
This sentence was combined in paragraph between lines 315-317 as follows:
In addition, PLpro showed deubiquitinase activity, which allows the virus to interrupt selective autophagy, preventing its proteins from being ubiquitinated (Mohamud et al., 2021).
- Line 289, "Another recent study" which one? Please cite.
The citation was added (Liang et al., 2021) in line 319.
The following reference was added:
“Liang S, Wu YS, Li DY, Tang JX, Liu HF. Autophagy in Viral Infection and Pathogenesis. Front Cell Dev Biol. 2021 Oct 15;9:766142. doi: 10.3389/fcell.2021.766142. PMID: 34722550; PMCID: PMC8554085 “
- Line 289-297, this part lacks citations.
Thank you for your comments we reformulate this paragraph and the corresponding citations were added in lines 319 and 324.
“SARS-CoV-2 uses autophagy to its benefit, hijacking the autophagy mechanism in the host cell to improve viral replication and to avoid the immune response and extracellular release. Viral proteins ORF3a and ORF7a were shown to cause accumulation of autophagosomes (Koepke et al., 2021). Particularly, ORF3a interacts with autophagy related protein UVRAG, suppressing autophagosome maturation and therefore the autophagy flux (Qu et al., 2021).”
The following references were added:
“Koepke, L.; Hirschenberger, M.; Hayn, M.; Kirchhoff, F.; Sparrer, K.M. Manipulation of autophagy by SARS-CoV-2 proteins. Autophagy., 2021;17(9):2659-2661. doi: 10.1080/15548627.2021.1953847.”
“Qu, Y.; Wang, X.; Zhu, Y.; Wang, W.; Wang, Y.; Hu, G. et al. ORF3a-Mediated Incomplete Autophagy Facilitates Severe Acute Respiratory Syndrome Coronavirus-2 Replication. Front Cell Dev Biol., 2021;9:716208. doi: 10.3389/fcell.2021.716208.”
- Line 373-375, Huang paper only reported the hyperinflammation caused by COVID-19, but they did not mention about autophagy. Inflammation is not always related to autophagy. So I think there is no adequate evidence to claim that SARS-CoV-2 induces inflammation via autophagy at this stage. Perhaps you can move this paragraph after later evide (or even in section 5).
Thank you for this comment. We are sorry for this mistake. As you suggested, in the revised version of our manuscript, we have moved the paragraph to Section 5 lines 451-454, where we think that it is more appropriate.
“Huang, in his prospective study, was the first to include an analysis of cytokine levels in severe and mild COVID-19, showing the presence of a cytokine storm analogous to that found for SARS-CoV infection (Huang et al., 2020). Excessive cytokine production could lead to multiorgan failure and worsen a patient's prognosis.”
- Line 404-408, are there any reports talking about SARS-CoV-2 regulating non-canonical pyroptosis through autophagy?
Thank you for this important comment. There is a very interesting paper from Sun et al, in Cell Death Diff 2022, in which they report that infection of cultured lung epithelial cells with live SARS-CoV-2 resulted in autophagic flux stagnation, inflammasome activation, and pyroptosis.
We address this observation in the revised version of our manuscript in lines 431-436 as follows:
“Sun et al. reported pyroptosis induction in patients with COVID-19. They found that NSP6 overexpression in SARS-CoV-2 infection may affect lysosome acidification in lung epithelial cells by interacting with vacuolar ATPase, leading to blockade of autophagic flux and inducing NLRP3 inflammasome activation and pyroptosis. When autophagy flux, is pharmacologically restore, NSP6-induced pyroptosis is suppressed (Sun X et al., 2022).”
We added the following reference:
Sun, X.; Liu, Y.; Huang, Z.; Xu, W.; Hu, W.; Yi, L. SARS-CoV-2 non-structural protein 6 triggers NLRP3-dependent pyroptosis by targeting ATP6AP1. Cell Death Differ., 2022 ;29(6):1240-1254. doi: 10.1038/s41418-021-00916-7.
- Line 448, please replace "COVID" with "SARS-CoV-2".
"COVID" was replaced by “SARS-CoV-2" in line 485.
- Line 443-451, this is an interesting point. Many prevalences of cardiovascular disease can increase the risk of COVID-19 symptoms. So is there any evi that hypertension also have an indencecreased NETosis?
This sentence was added in lines 485-488 regarding your comment:
“Moreover, it has been demonstrated that the increased level of angiotensin II seen in patients with hypertension could trigger NETosis, increasing the cardiovascular risk in these patients (Hofbauer et al., 2017; Chrysanthopoulou et al., 2021).”
The following references were added:
“Chrysanthopoulou A, Gkaliagkousi E, Lazaridis A, Arelaki S, Pateinakis P, Ntinopoulou M, Mitsios A, Antoniadou C, Argyriou C, Georgiadis GS, Papadopoulos V, Giatromanolaki A, Ritis K, Skendros P. Angiotensin II triggers release of neutrophil extracellular traps, linking thromboinflammation with essential hypertension. JCI Insight. 2021 Sep 22;6(18):e148668. doi: 10.1172/jci.insight.148668. PMID: 34324440; PMCID: PMC8492353.”
“Hofbauer, T., Scherz, T., Müller, J., Heidari, H., Staier, N., Panzenböck, A., ... & Lang, I. M. 2017. Arterial hypertension enhances neutrophil extracellular trap formation via an angiotensin-II-dependent pathway. Atherosclerosis, 263, e67-e68.”
- Line 464-466, please cite this part.
The citations in this part (Cicco et al., 2020 and Zuo et al., 2020) were added in the revised version of the manuscript in line 500.
The following references were added:
“Cicco, S.; Cicco, G.; Racanelli, V.; Vacca, A. Neutrophil Extracellular Traps (NETs) and Damage-Associated Molecular Patterns (DAMPs): Two Potential Targets for COVID-19 Treatment. Mediators Inflamm., 2020. 2020:7527953. doi: 10.1155/2020/7527953.”
“Zuo, Y.; Yalavarthi, S.; Shi, H.; Gockman, K.; Zuo, M.; Madison, J.A. et al. Neutrophil Extracellular Traps and Thrombosis in COVID-19. JCI Insight. 2020; 5(11):e138999. doi: 10.1172/jci.insight.138999.”
- Line 524, please verify whether the symptoms of "fever, fatigue malaise or systemic specific symptoms" are COVID-19 syndromes or post-COVID-19 syndromes.
Thank you very much for this comment. In the revised version of the manuscript, we added the following paragraph in section 6 (lines 563 to 565):
“The five most common symptoms are fatigue (58%), headache (44%), attention disorder (27%), hair loss (25%), and dyspnea (24%) (Lopez-Leon et al., 2021).”
The following reference was added:
“Lopez-Leon, S.; Wegman-Ostrosky, T.; Perelman, C.; Sepulveda, R.; Rebolledo, P.A.; Cuapio, A. et al. More than 50 long-term effects of COVID-19: a systematic review and meta-analysis. Sci Rep., 2021. 11(1):16144. doi: 10.1038/s41598-021-95565-8.”
- Line 550, I don't think you have ever mentioned about autophagy being involved in viral particle release.
Thank you very much for your comment, we apologize for this mistake and the line was eliminate in the revised manuscript. Line 591.
Autophagy would be involved in the entry as well as transcription and translation of the viral particles in the host cell (He et al. 2022).
Reviewer 2 Report
In this review, Resnik et al summarize the aspects that characterize the SARS-CoV-2 infections and autophagy. This manuscript is focused mainly in the viral replication cycle, the interplay among viral and cellular proteins related with autophagy, and then the authors are focused on pyroptosis and systemic inflammatory response.
The authors show three figures. The last figure is not mentioned in the manuscript at all.
I would like to suggest a general review of the English grammar to observe a more clear comprehensive review, together with a better structured version.
For specific points, please read below:
- Some paragraphs need attention. For example, line 46 should read : major histocompatibility complex (MHC). In lines 40-46 the authors talk about some ORfS proteins that will talk again later in another section.
- There is redundant information and repetitive sentences along the manuscript.
- Some references are missing in some of the sections. I.e. in line 54, line 88, line 237.
- In line 271 "it has been suggested that the virus enhances autophagy in other steps of this process" , but there is no further information about it.
- Line 380: "helps to clean the pathogens" is not an accurate sentence.
- the authors also describe mostly the anti viral role of the authophagy pathways, rather than the proviral role.
- The figure 1 is difficult to comprehend. Maybe highlighting the names of the different complexes along the pathway would help to clarify the steps.
Author Response
Comments and Suggestions for Authors
In this review, Resnik et al summarize the aspects that characterize the SARS-CoV-2 infections and autophagy. This manuscript is focused mainly in the viral replication cycle, the interplay among viral and cellular proteins related with autophagy, and then the authors are focused on pyroptosis and systemic inflammatory response.
We thank you very much for your comments, which allow us to significantly improve our manuscript. We have addressed them in the revised version of our manuscript and all the corrections are highlighted in green and listed below.
1-The authors show three figures. The last figure is not mentioned in the manuscript at all.
Thank you very much for this comment. We add lines 536 to 537 to introduce Figure 3 that now, in the revised version of our manuscript, is Figure 4.
“The interplay between local and systemic inflammatory response, SARS-CoV-2 and autophagy is summarized in figure 4.”
2-I would like to suggest a general review of the English grammar to observe a clearer comprehensive review, together with a better structured version.
Thank you for your suggestion. The new version of the manuscript was revised regarding structure, grammar, and comprehension. Also, we have submitted the revised version of our manuscript to the editing service offered by MDPI. Invoice ID: english-60773
For specific points, please read below:
3- Some paragraphs need attention. For example, line 46 should read: major histocompatibility complex (MHC).
Thank you, this was corrected in line 371 of the revised version of our manuscript.
4-In lines 40-46 the authors talk about some ORfS proteins that will talk again later in another section.
We deeply appreciate your comments and your help to improve our manuscript. Regarding your suggestion, the ORF proteins of SARS-CoV-2 play a crucial role in the interaction between the viral life cycle and the host cell. Therefore, in the revised version of our manuscript, we have introduced these proteins in the introduction (lines 43-49), but we described their functions in more detail in Section 3: “SARS-CoV-2 Infection and Its Effects on Autophagy” (lines 320-336) and (358-370)
Lines 43-49
“These reading frames generate eight accessory proteins: ORF3a, ORF3b, ORF6, ORF7a, ORF7b, ORF8a, ORF8b, and ORF9b (Yoshimoto et al., 2020; Lv et al., 2022). NSPs are key for viral RNA replication and immune avoidance, and the accessory proteins play diverse roles, aiding in viral infection, replication, and transmission (Thomas et al., 2021). Figure 1 depicts a diagram of the SARS-CoV-2 virus genome highlighting the two open reading frames: ORF1a and ORF1b (Lv et al., 2021)."
Figure 1: Genome of SARS-CoV-2 virus showing ORF1a and ORF1b open reading frame.
Lines 321-336
“Viral proteins ORF3a and ORF7a were shown to cause accumulation of autophagosomes (Koepke et al., 2021). Particularly, ORF3a interacts with autophagy related protein UVRAG, suppressing autophagosome maturation and therefore the autophagy flux (Qu et al., 2021). In two other studies, it was demonstrated that ORF3a interacts with VPS39 colocalizing with lysosomes. In this way, it impairs the binding of HOPS with RAB7, avoiding regulation of the fusion of autophagosomes with the lysosomes (Miao et al., 2021, Zhang et al., 2021). Another effect of viral protein ORF3a is its ability to promote lysosomal exocytosis, blocking autophagy flux and facilitating lysosomal targeting of the BORC-ARL8b complex. Additionally, BORC-ARL8b is involved in lysosomal trafficking and modulates the exocytosis-related SNARE complex (VAMP7, STX4, and SNAP23). Following this pathway, the complex is oriented towards the plasma membrane area. This entire process is Ca 2+ dependent (Chen et al., 2021). ORF7a generates a dysfunctional deacidified lysosome; therefore, autophagosomal degradation is interrupted and the virus can exit the host cell (Koepke et al., 2021).”
Lines 359-371
“Li and colleagues demonstrated a similar effect of ORF10 viral protein, which translocases to mitochondria and interacts with NIX—a protein very similar to the conforming protein of mitophagy receptor Nip3—and joins LC3 II. The activation of mitophagy leads to the degradation of mitochondrial antiviral signaling protein (MAVS), disrupting activation of type I INF. This suppresses cellular pyroptosis and cytokine release, hijacking the immune response in favor of SARS-CoV-2 survival (Li X et al., 2022). Additionally, SARS-CoV- 2-infected cells are much less sensitive to lysis by cytotoxic T lymphocytes. This could be due to non-structural viral protein ORF8, which impairs antigen presentation with major histocompatibility complex I (MHCI) and leads MHCI to lysosomal degradation, mediated via autophagy. This mechanism also helps to evade the immune response (Zhang et al., 2021). ORF8 also mediates the escape from the immune system by degrading major histocompatibility complex (MHC) (Flower et al., 2021).”
- 5 -There is redundant information and repetitive sentences along the manuscript.
Thank you for this comment. We apologize for this. In the revised version of our manuscript, we have rewritten several paragraphs that included redundant information. Please, see paragraphs highlighted in yellow.
- 6- Some references are missing in some of the sections. I.e. in line 54, line 88, line 237.
Thank you very much for this comment.
The citations in lines 54, 88 and 237 were added. In the revised version of the manuscript, they are in lines 75 (Kliche et al. 2021)., line 58 (Glick D et al. 2010) and line 221 (Schneider et al. 2021).
Moreover, as you suggested we completed references in all sections throughout the revised manuscript. We have added the following references in the following lines of the revised version of our manuscript:
Line 63 (Dahmane et al., 2022):
Dahmane, S.; Shankar, K.; Carlson, L.A. A 3D view of how enteroviruses hijack autophagy. Autophagy. 2022 Dec 5:1-3. doi: 10.1080/15548627.2022.2153572. Epub ahead of print. PMID: 36471479.
Line 488 (Chrysanthopoulou et al., 2021):
Chrysanthopoulou, A.; Gkaliagkousi, E.; Lazaridis, A.; Arelaki, S.; Pateinakis, P.; Ntinopoulou, M. et al. Angiotensin II triggers release of neutrophil extracellular traps, linking thromboinflammation with essential hypertension. JCI Insight. 2021 Sep 22;6(18):e148668. doi: 10.1172/jci.insight.148668. PMID: 34324440; PMCID: PMC8492353.
Line 488 (Hofbauer et al., 2017):
Hofbauer, T.; Scherz, T.; Müller, J.; Heidari, H.; Staier, N.; Panzenböck, A. et al. Arterial hypertension enhances neutrophil extracellular trap formation via an angiotensin-II-dependent pathway. 2017, Atherosclerosis, 263, e67-e68.
Line 352 (Sun et al., 2022):
Sun, X.; Liu, Y.; Huang, Z.; Xu, W.; Hu, W.; Yi, L. SARS-CoV-2 non-structural protein 6 triggers NLRP3-dependent pyroptosis by targeting ATP6AP1. Cell Death Differ., 2022 ;29(6):1240-1254. doi: 10.1038/s41418-021-00916-7.
Line 319 (Liang et al., 2021):
Liang S, Wu YS, Li DY, Tang JX, Liu HF. Autophagy in Viral Infection and Pathogenesis. Front Cell Dev Biol. 2021 Oct 15;9:766142. doi: 10.3389/fcell.2021.766142. PMID: 34722550; PMCID: PMC8554085.
Line 49 (Lv et al., 2022):
Lv, Z.; Cano, K.E.; Jia, L.; Drag, M.; Huang, T.T.; Olsen, S.K. Targeting SARS-CoV-2 Proteases for COVID-19 Antiviral Development. Front Chem. 2022 Feb 3;9:819165. doi: 10.3389/fchem.2021.819165. PMID: 35186898; PMCID: PMC8850931.
Line 322 ( Koepke et al., 2021)
Koepke, L.; Hirschenberger, M.; Hayn, M.; Kirchhoff, F.; Sparrer, K.M. Manipulation of autophagy by SARS-CoV-2 proteins. Autophagy., 2021;17(9):2659-2661. doi: 10.1080/15548627.2021.1953847.
Line 197 (Sargazi et al., 2021)
Sargazi, S.; Sheervalilou, R.; Rokni, M.; Shirvaliloo, M.; Shahraki, O.; Rezaei, N. The role of autophagy in controlling SARS-CoV-2 infection: An overview on virophagy-mediated molecular drug targets. Cell Biol Int. 2021 Aug;45(8):1599-1612. doi: 10.1002/cbin.11609. Epub 2021 Apr 23. PMID: 33818861; PMCID: PMC8251464.
Line 152 (Hama et al., 2022)
Hama, Y.; Morishita, H.; Mizushima, N. Regulation of ER-derived membrane dynamics by the DedA domain-containing proteins VMP1 and TMEM41B. EMBO Rep. 2022 Feb 3;23(2):e53894. doi: 10.15252/embr.202153894
Line 228 (Trimarco et al., 2021)
Trimarco, J.D.; Heaton, B.E.; Chaparian, R.R.; Burke, K.N.; Binder, R.A.; Gray, G.C. et al. TMEM41B is a host factor required for the replication of diverse coronaviruses including SARS-CoV-2. PLoS Pathog. 2021 May 27;17(5):e1009599. doi: 10.1371/journal.ppat.1009599. PMID: 34043740; PMCID: PMC8189496.
Line 239 (Scutigliani et al., 2017)
Scutigliani, E.M.; Kikkert, M. Interaction of the innate immune system with positive-strand RNA virus replication organelles. Cytokine Growth Factor Rev. 2017 Oct;37:17-27. doi: 10.1016/j.cytogfr.2017.05.007. Epub 2017 Jun 27. PMID: 28709747; PMCID: PMC7108334.
Line 235 (Snijder et al., 2020)
Snijder ,E.J.; Limpens, R.W.A.L.; de Wilde, A.H.; de Jong, A.W.M.; Zevenhoven-Dobbe, J.C.; Maier, H.J. et al. A unifying structural and functional model of the coronavirus replication organelle: Tracking down RNA synthesis. PLoS Biol. 2020 Jun 8;18(6):e3000715. doi: 10.1371/journal.pbio.3000715. PMID: 32511245; PMCID: PMC7302735
Line 228 (Ji et al., 2022)
Ji, M.; Li, M.; Sun, L.; Zhao, H.; Li, Y.; Zhou, L. et al. VMP1 and TMEM41B are essential for DMV formation during β-coronavirus infection. J Cell Biol. 2022 Jun 6;221(6):e202112081. doi: 10.1083/jcb.202112081. Epub 2022 May 10. PMID: 35536318; PMCID: PMC9097365.
Line 365 (Li, X et al., 2022)
Li, X.; Hou, P.; Ma, W.; Wang, X.; Wang, H.; Yu, Z. et al. SARS-CoV-2 ORF10 suppresses the antiviral innate immune response by degrading MAVS through mitophagy. Cell Mol Immunol., 2022. 19(1):67-78. doi: 10.1038/s41423-021-00807-4.
Line 301 ((Li, F et al., 2021)
“Li, F.; Li, J.; Wang, P.H.; Yang, N.; Huang, J.; Ou, J. et al. SARS-CoV-2 spike promotes inflammation and apoptosis through autophagy by ROS-suppressed PI3K/AKT/mTOR signaling. Biochim Biophys Acta Mol Basis Dis. 2021 Dec 1;1867(12):166260. doi: 10.1016/j.bbadis.2021.166260. Epub 2021 Aug 27. PMID: 34461258; PMCID: PMC8390448.”
-7 In line 271 "it has been suggested that the virus enhances autophagy in other steps of this process" , but there is no further information about it.
We have addressed this comment by adding the following paragraph in line 295-301 of the revised version of our manuscript.
“Li et al. explored the regulatory role of the SARS-CoV-2 spike protein in infected cells and attempted to elucidate the molecular mechanism of SARS-CoV-2-induced inflammation. They found that SARS-CoV-2 inhibits the PI3K/AKT/mTOR pathway by upregulating intracellular reactive oxygen species (ROS) levels, and, in this way, promotes the autophagic response. Subsequently, SARS-CoV-2-induced autophagy triggers inflammatory responses and apoptosis in infected cells (Li et al., 2021).”
The following reference was added:
“Li, F.; Li, J.; Wang, P.H.; Yang, N.; Huang, J.; Ou, J. et al. SARS-CoV-2 spike promotes inflammation and apoptosis through autophagy by ROS-suppressed PI3K/AKT/mTOR signaling. Biochim Biophys Acta Mol Basis Dis. 2021 Dec 1;1867(12):166260. doi: 10.1016/j.bbadis.2021.166260. Epub 2021 Aug 27. PMID: 34461258; PMCID: PMC8390448.”
- 8 Line 380: "helps to clean the pathogens" is not an accurate sentence.
Thank you for your observations and your help to improve our manuscript. The sentence in line 404 was changed using proper grammar to
“…contributes to pathogen clearing.”
- 9-the authors also describe mostly the anti-viral role of the autophagy pathways, rather than the proviral role.
Thank you for your observations. Pro-viral role is described in section: 2 “Autophagy and the viral replication cycle”.
Lines 210 to 213:
“Cathepsin L, which links the virus cycle to the autophagy process, acts at low pH, degrades cargo, and maintains autolysosome homeostasis and autophagic flux (Xu et al., 2021). By cleaving the spike protein at S2´, it mediates virus membrane and autolysosome fusion, thus facilitating the release of viral RNA into the host cell.”
Lines 215 to 217:
“Furin, and Cathepsin L proteases have cumulative effects to activate virus entry and increase the pathogenicity of SARS-CoV-2 (Schneider et al., 2021; Qu et al., 2021). “
Lines 220 to 228:
“Recently, as shown in Figure 3, pathway 3, transmembrane proteins related to autophagy VMP1 and TEMEM41B have been reported as critical host factors at the early stages of viral infection (Schneider et al., 2021). VMP1 and TMEM41B both contribute to different stages of DMV formation. Ji et al. have revealed that DMV biogenesis is impaired in VMP1 and TMEM41B knockout cells. Analysis using transmission electron microscopy revealed that the formation of DMVs was substantially inhibited in cells lacking these autophagy proteins. Hence, by inhibiting VMP1 and TMEM41B expression, the virus is unable to hide from the immune sensors in DMVs, thus reducing its protection from the immune system (Ji et al., 2022; Trimarco et al., 2021).”
Lines 249 to 264:
“Another connecting pathway between the viral replication cycle and autophagy is represented by SNX27, one of the sorting nexins (SNX) family members, which down-regulates autophagy by increasing the level of mTORC1 signaling (Yang Z. et al., 2018). Figure 3, pathway 2 shows how SNX27 regulates the traffic of endosomal receptors towards recycling endosomes. Kim et al. found that mTORC1 acts as a signal integrator at the lysosome and can act as an inhibitor of later stages of autophagy, suppressing phosphorylation on UVRAG, which is a component of VPS 34 complex II. In this way, it avoids autophagosome and endosome maturation (Kim et al., 2018). These events are relevant for the viral cycle, given that the virus enters the cell via directly fusing to the membranes in the cell surface pathway or via the endocytic pathway through endosome/lysosome-mediated endocytosis. Interestingly, it has recently been found that the ACE2 receptor possesses a type I PDZ binding motif (PBM) and can therefore interact with a PDZ domain-containing protein such as SNX27. A recent study has shown SNX27 to be critical for ACE2 cell surface regulation, and SNX27 prevents ACE2-bonded viral particles from entering the lysosome, down-regulating the endocytic viral entry pathway, and therefore serving as a viral trafficking regulator (Yang B. et al., 2022).”
In addition, we include further information on this topic in the revised manuscript in the following paragraph (lines 194-197)
“Several viruses, including coronaviruses (CoVs), take advantage of cellular autophagy to facilitate their own replication. SARS-CoV-2 mediates its replication through a dependent or independent ATG5 pathway using specific double-membrane vesicles that can be considered similar to autophagosomes (Sargazi et al., 2021).”
We added the following reference:
“Sargazi, S.; Sheervalilou, R.; Rokni, M.; Shirvaliloo, M.; Shahraki, O.; Rezaei, N. The role of autophagy in controlling SARS-CoV-2 infection: An overview on virophagy-mediated molecular drug targets. Cell Biol Int. 2021 Aug;45(8):1599-1612. doi: 10.1002/cbin.11609. Epub 2021 Apr 23. PMID: 33818861; PMCID: PMC8251464.”
10 - The figure 1 is difficult to comprehend. Maybe highlighting the names of the different complexes along the pathway would help to clarify the steps.
Thank you for your observation. We have revised Figure 1 (Figure 2 in the revised version of our manuscript) to include a clearer representation of the complexes and their names. Additionally, a list of the complexes has been added to the legend of this figure.
The corresponding modification is in lines 135-137.
“The complexes pointed out in this figure are: complex ULK1 (ULK1-ATG13-FIP200-ATG10); complex class III PI3-kinase: (BECN1-ATG14L-VPS34-VPS15); complex PI3KC3-C2: (BECN1-UVRAG-VPS34-VPS15).”
And in lines 140-142
“The UKL1 complex, ULK1-ATG13-FIP200-ATG101, drives the pre-autophagosomal structure. Then, the Class III PI3K-kinase complex,…”
Round 2
Reviewer 1 Report
Thanks for your revision. All my concerns are addressed.
Reviewer 2 Report
In this review of the manuscript, the authors clarified all the points asked in the related work.